# Image Process of Rock Size Distribution Using DexiNed-Based Neural Network

**Haijie Li [1,*], Gauti Asbjörnsson [2] and Mats Lindqvist [1]**

[1]  Mining R&D, FLSmidth A/S, 2500 Valby, Denmark; mats.lindqvist@flsmidth.com
[2]  Department of Industrial and Materials Science, Chalmers University of Technology, 412 96 Gothenburg, Sweden; gauti@chalmers.se
*  Correspondence: hli@flsmidth.com

**Abstract:** In an aggregate crushing plant, the crusher performances will be affected by the variation from the incoming feed size distribution. Collecting accurate measurements of the size distribution on the conveyors can help both operators and control systems to make the right decisions in order to reduce overall power consumption and avoid undesirable operating conditions. In this work, a particle size distribution estimation method based on a DexiNed edge detection network, followed by the application of contour optimization, is proposed. The proposed framework was carried out in the four main steps. The first step, after image preprocessing, was to utilize a modified DexiNed convolutional neural network to predict the edge map of the rock image. Next, morphological transformation and watershed transformation from the OpenCV library were applied. Then, in the last step, the mass distribution was estimated from the pixel contour area. The accuracy and efficiency of the DexiNed method were demonstrated by comparing it with the ground-truth segmentation. The PSD estimation was validated with the laboratory screened rock samples.

**Keywords:** image processing; image segmentation; particle size distribution; OpenCV; convolutional neural networks; DexiNed





## 1. Introduction

In a comminution circuit, it is desirable to have an accurate knowledge of the particle size in various process steps. Nowadays, most mineral and crushing plants detect material size by manual screening, which involves enormous human labor resources. With the rapid development of artificial intelligence, advanced image processing technology and machine vision have been increasingly utilized in particle size detection. Hence, image analysis provides the opportunity to gauge the size of rock particles coming out of a crushing process.

Many researchers have developed particle size detection methods based on traditional image processing techniques and showed significant breakthroughs. These methods are mainly developed based on watershed-related models [1,2] and threshold segmentation [3,4]. However, these traditional image processing methods rely on dedicated parameter tuning and are difficult to be generalized to other scenarios [5].

With the advancement in convolutional neural networks and machine learnings, machine learning-based segmentation methods have attracted great attention [6]. Mukherjee et al. [7] developed a neural network to enhance the uneven light images and to learn rock shape features. Yuan et al. [8] used a deep learning method to solve the mutual adhesion and shadow problems in the rock images. Besides the generalized neural network structures, the convolutional neural network (CNN)-based algorithms present significant advantages in object detection and semantic segmentation [9]. Ma et al. [10] proposed a CNN-based method to firstly identify the conveyor belt situation, and then to choose between fine rock filter and coarse rock filter.

In 2015, the U-net structure was proposed by Ronneberger et al. [11] for biomedical image segmentation. Since then, U-net and U-net variants became the benchmarks in

many medical-related segmentation tasks. In ore segmentation, Liu et al. [12] adopted U-net and combined it with Res_Unet for the conveyor belt ore image segmentation. They compared their results with a single U-net and a watershed algorithm. It demonstrated a better accuracy of their proposed two-step contour detection model. Wang et al. [5] used a different decoder in U-net to enhance the boundary awareness. Similarly, Yang et al. [6] improved the encoder-decoder based on a U-Net to deal with low contrast and blurry boundaries of ores. However, these segmentation models generally need massive labeled training data. Therefore, an edge detection algorithm that does not need an extensive training process would be more efficient and computationally inexpensive for the rock segmentation task.

DexiNed is a CNN-based edge detection algorithm proposed by Poma et al. [13]. The architecture of DexiNed is influenced by Xception and ResNet. Each of the main blocks in the structure is composed of sub-blocks and densely interconnected by the previous main block. The training process of DexiNed is end-to-end without excessive manual tuning [14]. Poma et al. pointed out that even without a pre-trained dataset, DexiNed still outperformed the state-of-the-art edge detection algorithms [13].

Taking the advantages of DexiNed, the rock edge maps in this work are predicted without any pre-trained process. The main contributions of this study include:

- Using a modified DexiNed network to detect the edge maps of rock images;
- Propose processes to separate fine regions, connected neighboring rocks, and further segment the connected material into individual particles;
- Convert the pixel areas detected from drawn contours to mass distribution and validate results with the lab screening data.

## 2. Materials and Methods

The proposed method can be divided into four main steps, see Figure 1. Firstly, the raw input images are preprocessed with gray scaling and contrast-limited adaptive histogram equalization (CLAHE). The preprocessed images have higher contrast, sharper edges, and even light conditions. Then a DexiNed-based network is applied to predict the thin edge maps. Furthermore, the edge maps need to be processed by binarization and morphological transformations. In this step, the particle contours are classified into fine particles, adjacent rocks, and individual rocks. Each adjacent rock contour is separated into smaller segmentation areas. Finally, the pixel areas of rock fragments are collected and calculated to generate the particle size distribution of the given image.

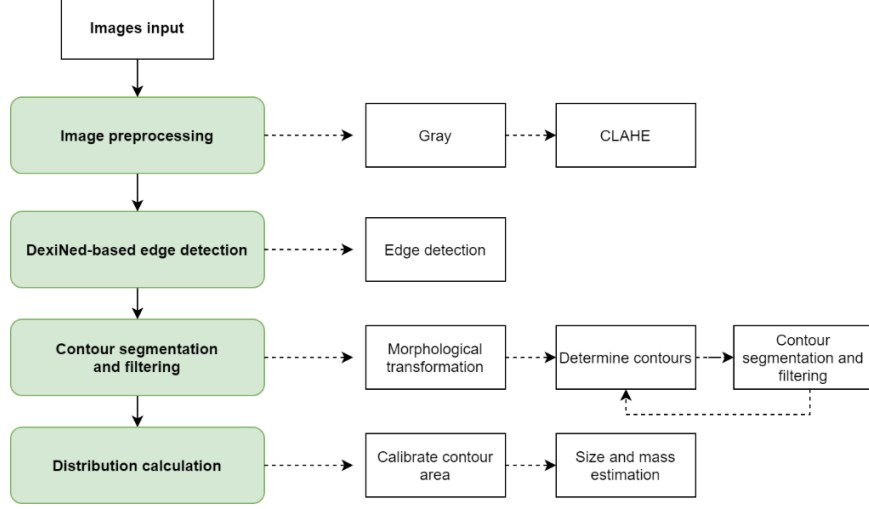

**Figure 1.** A flowchart of the proposed processes of ore size distribution estimation.

### 2.1. Data Preparation and Image Preprocessing

The DexiNed edge detection can be fed with unprocessed raw RGB images and gives good edge predictions without pre-trained data. The gray scaling and CLAHE are applied to make the neural network more efficient and generate more explicit thin-edge maps. The uneven color and light are mitigated, as shown in Figure 2b,c.

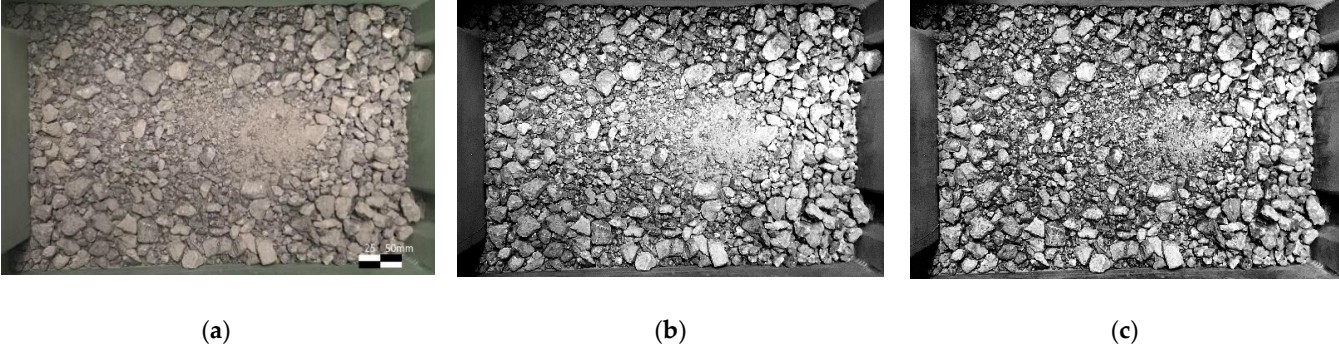

(**a**)            (**b**)            (**c**)

**Figure 2.** An example of a raw image from lab sampling and its preprocessed results: (**a**) the raw image; (**b**) the grayscale image; (**c**) the contrast-limited adaptive histogram equalization (CLAHE) result.

The tested rock images are generated from a laboratory rock sample that is taken from cone crusher product with sizes ranging from 0 to 22.4 mm. The sample was sieved to obtain its actual distribution for validation of the results of the proposed method.

The DexiNed network can be used without the pre-trained datasets, so the experiment is carried out without the training process. The sample was shuffled and photographed to obtain sample images of the same material. Five representative images of the same rock sample were selected for further analysis and final evaluation.

### 2.2. DexiNed-Based Boundary Detection

The DexiNed network can be fed with the RGB images of various objects and predicts a thin-edge map with the same resolution. Adopted from the original DexiNed architecture, a modified DexiNed is used in this work. Considering the preprocessing of rock images and the similarities of rock edge features, the main blocks are reduced to five instead of six in the original algorithms. This avoids the loss of details in the final fused result.

The modified DexiNed structure has two main components, as shown in Figure 3: Dense extreme inception (Dexi) block and the up-sampling block. The up-sampling block is fed with outputs from the Dexi blocks, while each Dexi block is fed with previous image features.

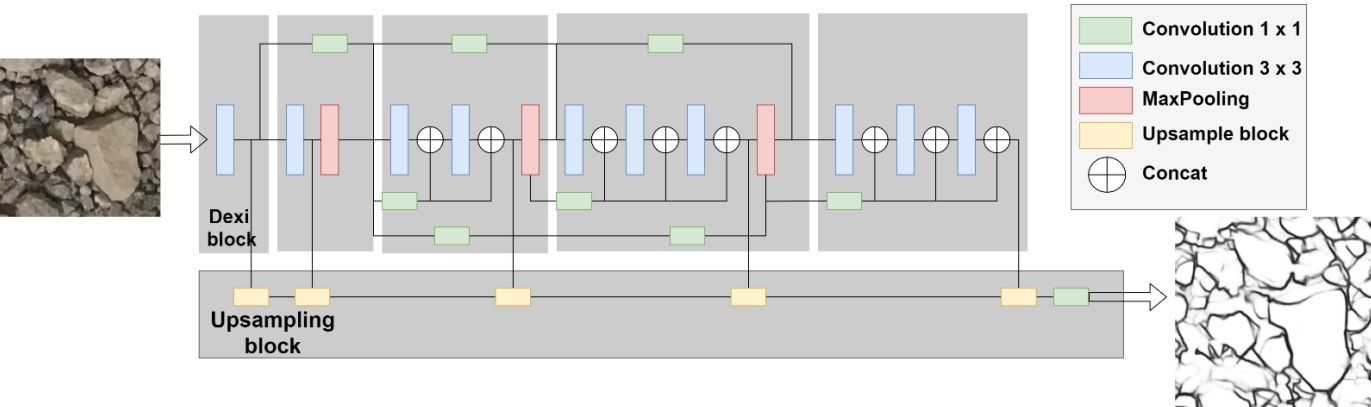

**Figure 3.** The network structure of DexiNed-based edge prediction model.

The edge map of the rock regions is obtained from the convolutional networks, as seen in Figure 4. The predicted contours need to be further processed to deal with boundary discontinuity or over-segmentation in the fused edge map. To improve the accuracy of rock image segmentation, more filters based on the contour area, hierarchy, and shapes were applied.

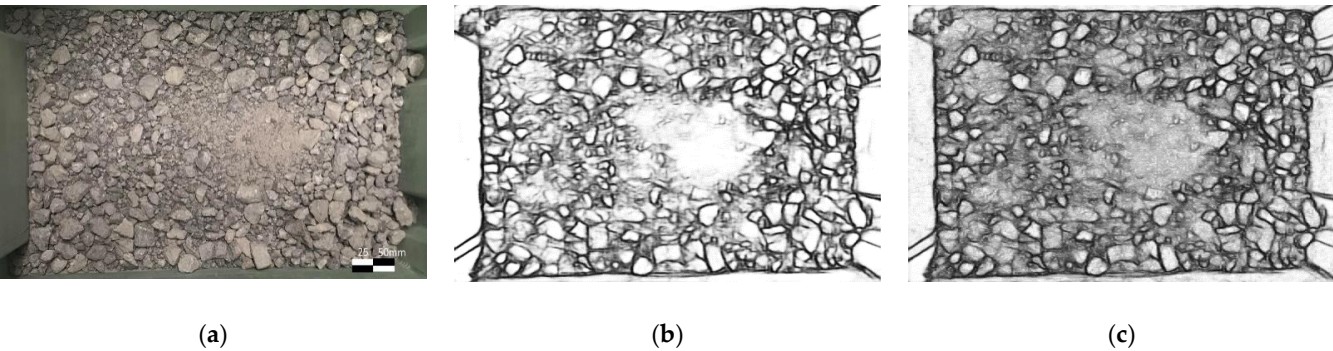

(a)          (b)          (c)

**Figure 4.** An example of raw image and its edge detection results: (**a**) the raw image; (**b**) the fused edge-map concatenated from upsampling blocks; (**c**) the averaged edge-map from the previous predictions.

### 2.3. Segmentation Process

The details of rock segmentation processes from the DexiNed predicted-edge map are given as follows, and the example results are displayed in Figure 5. First, we apply a binary filter on fused-edge maps. The threshold value is set to 180 in the OpenCV binary threshold function. Then, the morphological operation is employed on the binary image, where a kernel window of size (3 by 3) scans the whole image. Erosion is carried out for this morphological operation. The erosion function in OpenCV erodes away the boundaries of foreground object. This process reduces the connected area between neighboring rocks. The iteration = 1 and the eroded binary image is shown in Figure 5b. The reduced area caused by the morphological operators will be compensated in a later process.

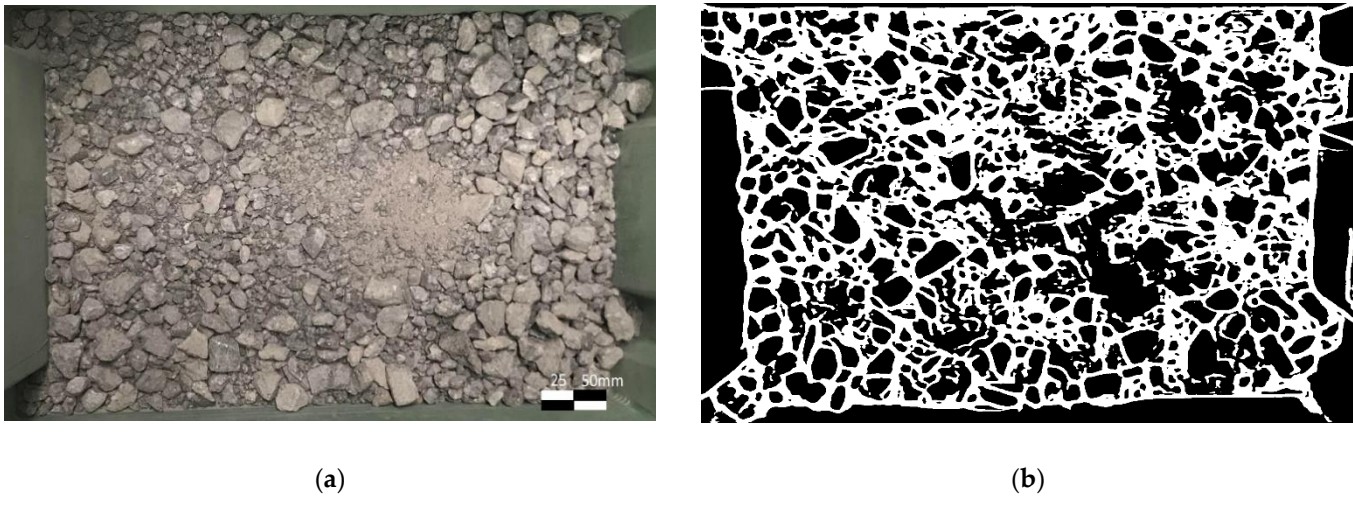

(a)          (b)

**Figure 5.** *Cont.*

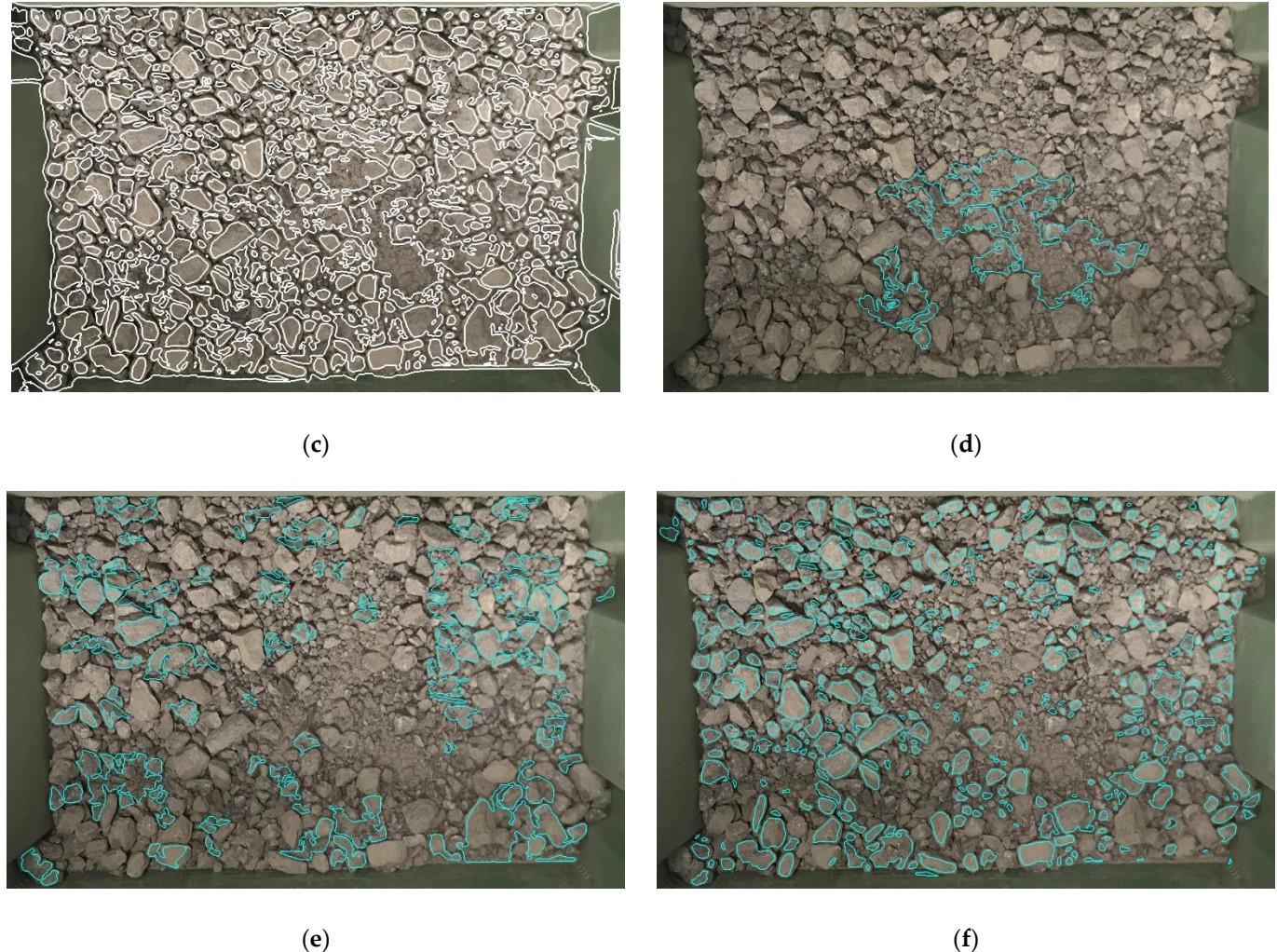

**Figure 5.** An example of raw image and processed results: (**a**) the raw image; (**b**) the binary image of detected edge; (**c**) all contours found; (**d**) fine region; (**e**) connected rock region; (**f**) single rock region.

Next, we apply OpenCV 'ContourArea', this function returns the pixel area of an enclosed region. The convex area for each contour can be determined by the function 'ConvexHull', we use the ratio between the convex hull area and the contour area to filter the single rocks from fine particles and neighboring rocks. Figure 5c illustrates all the contours identified by the OpvenCV built-in function 'findContours'. The ratio between the convex hull area and the contour area is obtained:

$$R_c = A_c / A_h \tag{1}$$

where $R_c$ is the area ratio, $A_c$ is the contour area, and $A_h$ is the convex hull area. $R_c$ = 1.2 is the threshold to determine if this contour belongs to one single rock or several connected rocks. In this step, we set an upper limit for the pixel area where a region larger than the upper limit is least likely a single rock. $R_c$ also needs to be considered. Then we obtain the hierarchy for each contour that helps to remove the sub contours.

After the previous filtering processes, all the remaining contours are categorized into three classes: the fine region in Figure 5d; the connected rocks in Figure 5e; the single rock in Figure 5f.

Now, the connected rock areas are examined, respectively. As shown in Figure 6, the minimum rectangle area of each adjacent rock is extracted. The small images then go through the binary threshold function and morphological operation again. The binary threshold value increases from 180 to 250. The contour change for each iteration is compared,

as shown in Figures 7 and 8, and the optimal threshold value for extracted contour map is selected. It can be seen in Figure 7, the contours labeled by blue lines shrink as the threshold value increases. The image is divided into more non-connected regions at value = 200 and value = 250, respectively. For this specific example, the new threshold value should be 200, as a larger threshold value results in underestimating the rock area. Figure 8 shows the total number of particle contours and total pixel area. As expected, the total pixel area declines while the contour number reaches a maximum at value = 210. The global threshold value = 210 is then selected for separating the single rocks from neighboring rock regions.

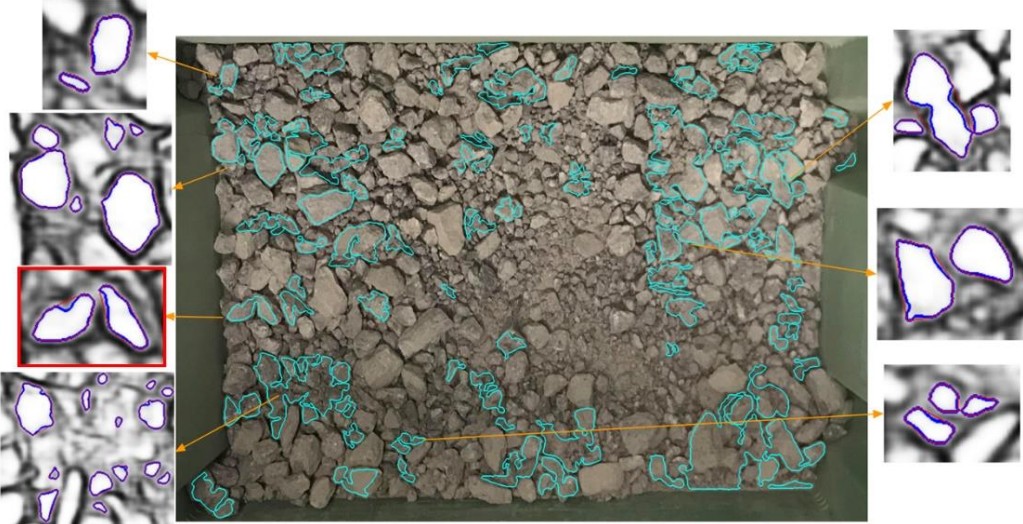

**Figure 6.** An example of the connected neighboring rocks and the contour map of the extracted images with the new threshold value. The highlighted connected rock region (red box) and its separation process is presented in Figure 7.

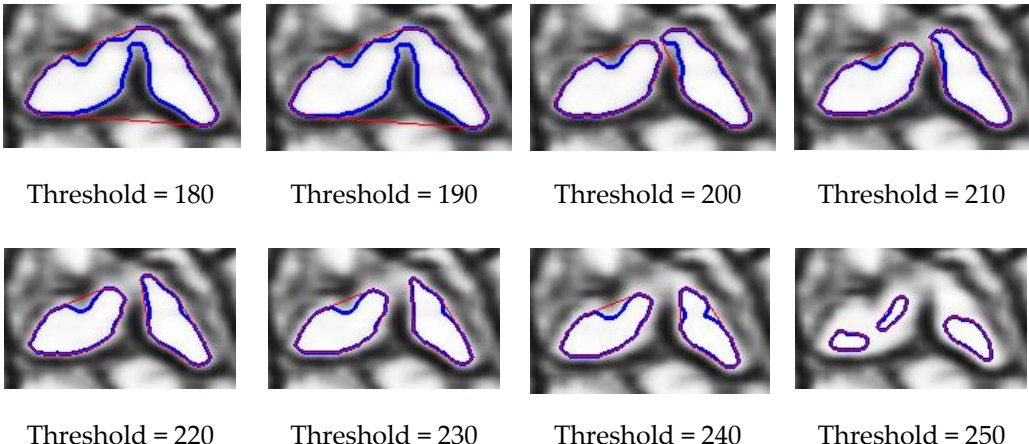

| Threshold = 180 | Threshold = 190 | Threshold = 200 | Threshold = 210 |
| Threshold = 220 | Threshold = 230 | Threshold = 240 | Threshold = 250 |

**Figure 7.** The separation process of an extracted image with two neighboring rocks. The contours are shown with blue lines and the convex hulls are shown by red lines.

Once the contours are drawn, the following equations are utilized to compensate for the area loss caused by the morphological operation. This calibration has a significant impact on the small rocks.

$$A_{all} = A_c + A_a \tag{2}$$

$$A_a = C_c * S_k \tag{3}$$

where $A_{all}$ is the total contour area that is identified, $A_c$ is a list of the contour area obtained via the 'findContour' function from all the labeled contours, and $A_a$ is a list of adjusted areas. In Equation (3), $A_a$ is calculated from the contour perimeter $C_c$ via the 'ArcLength' function, and $S_k$ is the kernel window size. A morphological transformation has been applied to have a better segmentation of edge maps. Thus, the corresponding area shrinks along the edges. The compensation area for each region is approximately the product of its perimeter and kernel window size.

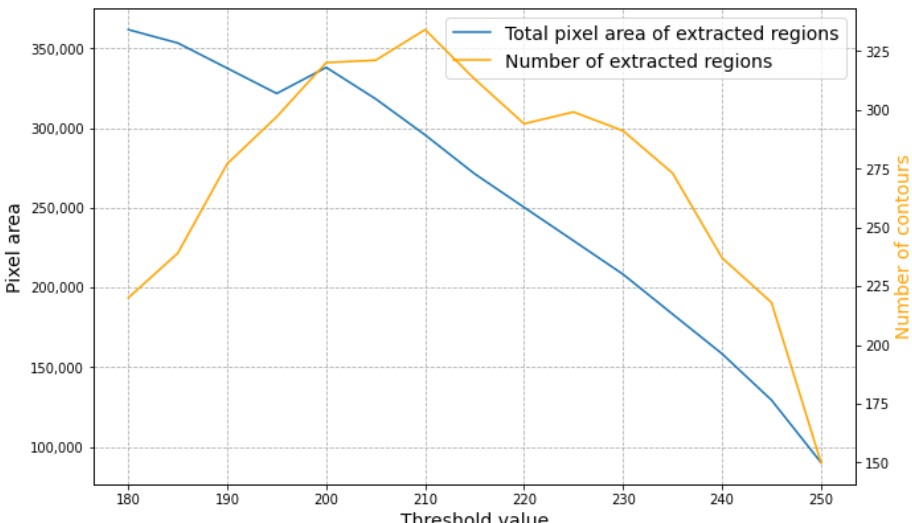

**Figure 8.** An example of the separation process of the connected rock regions. The new threshold value is chosen based on the total pixel area (blue line) and number of extracted regions (orange line).

The scenario of this study is the rock segmentation for feed/product of cone crushers. Only the non-overlapped rocks are identified in the process because the overlapped rocks are mostly filtered:

- Due to the uneven light conditions, even with the CLAHE enhancement, it is difficult to draw clear edges in the shadow;
- Binary threshold and morphological transformation remove background noises, see Figure 5b;
- The shape of the rocks is assumed convex while the visible part of overlapped rocks could be concave. So overlapped rocks are filtered by area ratio $R_c$;
- The hierarchy examination of each contour removes the children regions and parent regions.

## 3. Experimental Results and Discussion

This study is carried out on the Google Colab pro, Python 3.7.10, and OpenCV version 4.1.2.30. The image acquisition device is Apple iPhone X rear camera. The original image resolution is 4032 by 3024. Images are chopped to fit the boundaries of the samples. The box size is 345 mm by 490 mm. Five images are selected after totally reshuffling the samples. The average execution time per image is 30 ms for the DexiNed edge mapping, 200 ms for the contour segmentation and 50 ms for the mass distribution estimation.

### 3.1. Performance Evaluation

The pixel areas of the closed contours are counted, and the cumulative/frequency area distribution is calculated. The area screen filters are set to be 2000, 4000, 6000, 8000, 10,000, and larger than 10,000 pixels. Due to the rock sample size (0–22.4 mm), it is not feasible to label all the rocks in an image manually, especially with fine regions. Hence, the comparison between the manual label image and the auto-processed images is not considered an evaluation criterion but serves as a demonstration. Instead, the mass is

estimated from the pixel area distributions. The conversion from the pixel area to mass is conducted with the following equation:

$$M_c = F_{scale} \times A_{all}^{\frac{3}{2}} / R_{img}^3 \qquad (4)$$

$$M_f = F_{scale} \times A_f \times T / R_{img}^3 \qquad (5)$$

where $M_c$ is the list of the coarse particles mass and $R_{img}$ (px/mm) is the pixel ratio to the real dimensions. $R_{img}$ varies from 35 to 39 in different images. For example, the box dimension is 1217 by 1896 px in image 3. The corresponding pixel ratio is thus $R_{img} = 37$. It should be noted that this is a rough estimation that assumes a coarse rock as a sphere without considering the bulk density, and then it is scaled to fit the measured mass using $F_{scale}$. $F_{scale}$ is determined from the particles weight and its corresponding pixel area, in this study, $F_{scale} = 2.5$. $M_f$ is the mass of fine particles and is calculated from the fine contour area $A_f$ and the thickness of the fine layer is set $T = 15$ mm.

### 3.2. Result Analysis of Area and Mass Distribution

Images from Figure 9 show the fused-edge maps and filtered results of the fine region, the connected rock region, and the single rock region, respectively. The connected rock regions are processed again with adjusted threshold values. The details are explained in Section 2.3.

The calculated contour area distributions can be found in Tables 1 and 2. For the illustration purpose, the manual label of image 1 is presented, see Figure 10. The white rectangle shows a pixel area of 2000. It is assumed that all coarse rocks larger than 2000 px are labeled. Therefore, the total contour area differences between manual and auto processes are categorized into the 0–2000 px range.

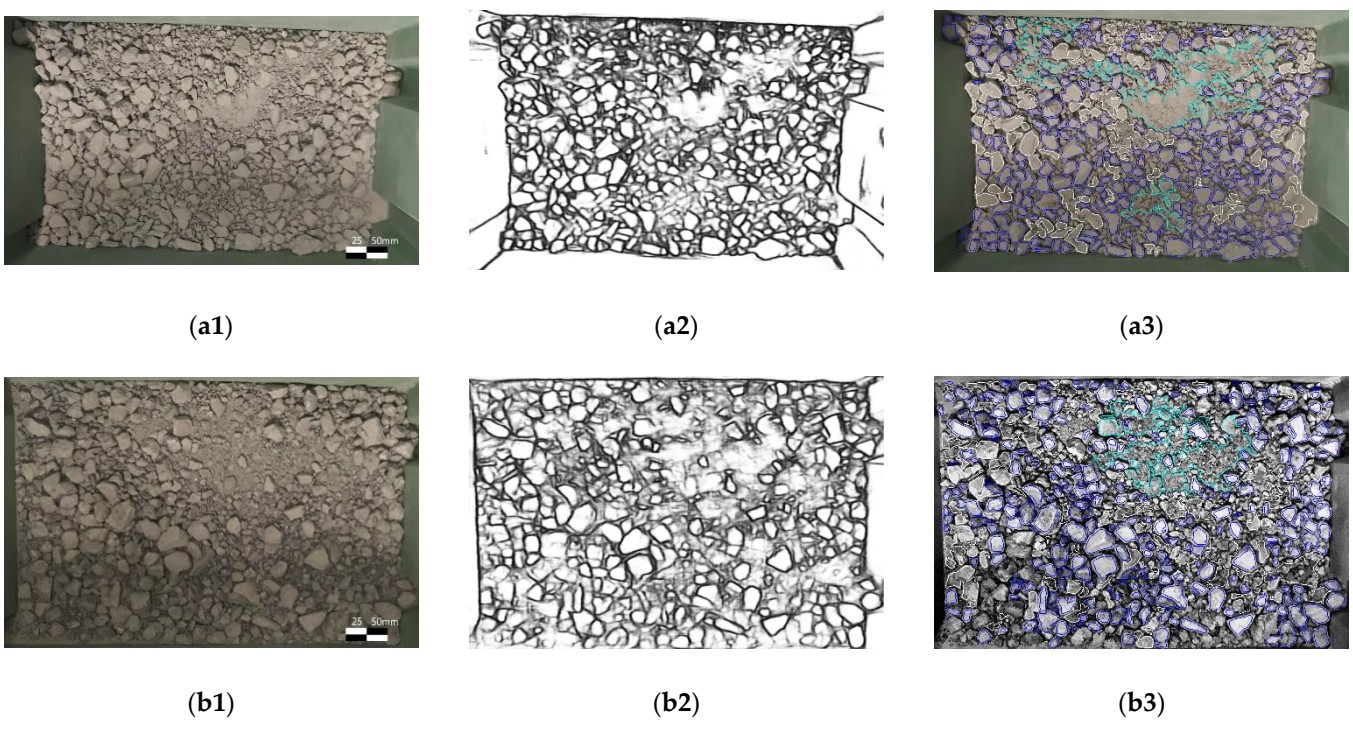

| (a1) | (a2) | (a3) |
| --- | --- | --- |
| (b1) | (b2) | (b3) |

**Figure 9.** *Cont.*

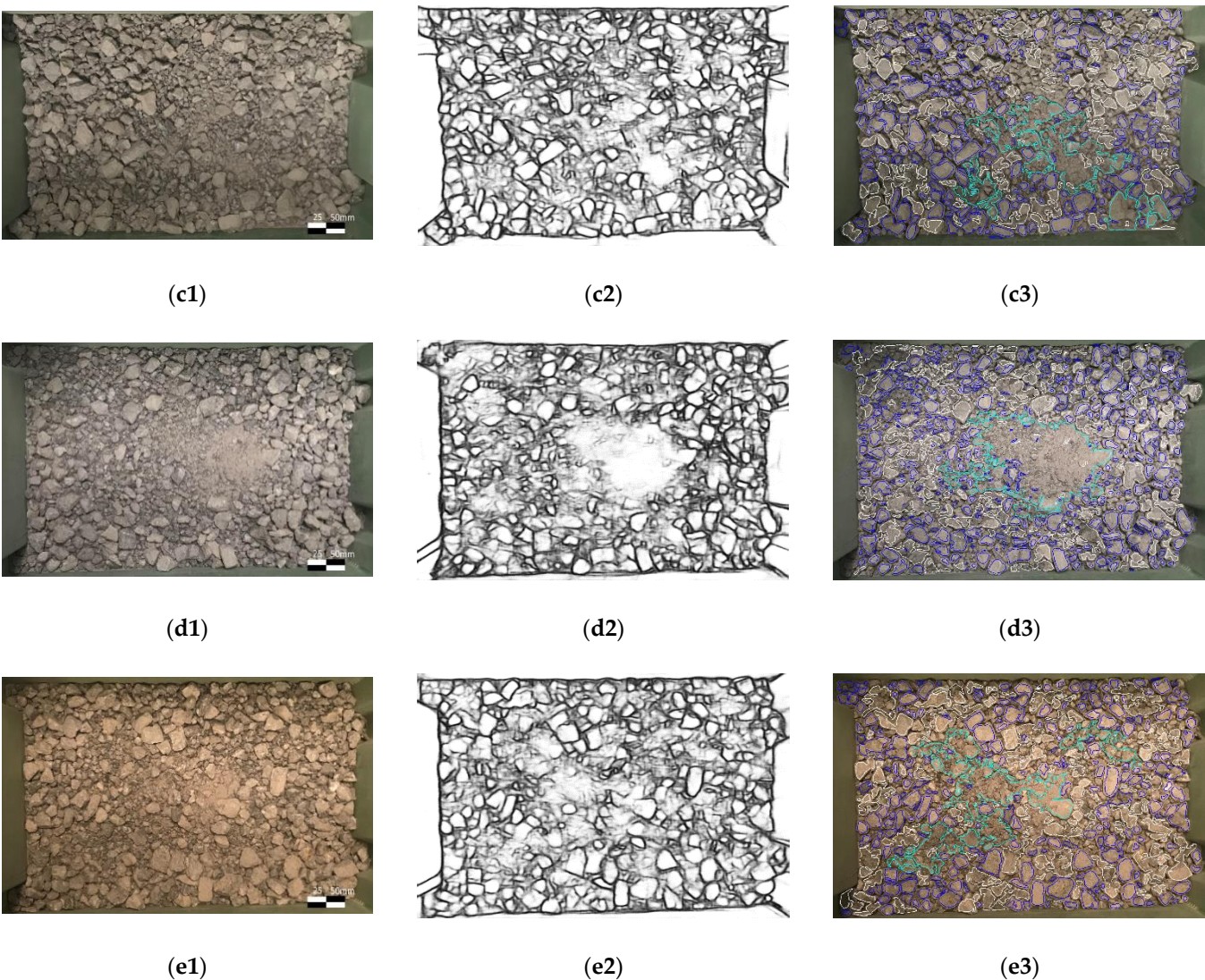

**Figure 9.** The processed results: (**a1,b1,c1,d1,e1**) raw images of lab samples; (**a2,b2,c2,d2,e2**) edge maps of input images; (**a3,b3,c3,d3,e3**) contour maps with three filters results, where the light green contour is the fine region, the white contour is the connected rock region and the blue is single rock region.

**Table 1.** The frequency of rock pixel area distribution from images 1 to 5, compared with manual results of image 1.

| Image No. | Frequency of Pixel Area Distribution (%) | | | | | |
|---|---|---|---|---|---|---|
| | 2000 | 4000 | 6000 | 8000 | 10,000 | >10,000 |
| Image 1 | 43.4 | 25.9 | 15.6 | 8.5 | 4.4 | 2.2 |
| Image 2 | 40.2 | 26.8 | 15.7 | 10.5 | 4.8 | 2.0 |
| Image 3 | 43.9 | 24.3 | 11.8 | 10.9 | 3.4 | 5.8 |
| Image 4 | 45.2 | 26.8 | 10.9 | 10.0 | 1.4 | 5.7 |
| Image 5 | 41.9 | 23.0 | 13.1 | 9.5 | 7.0 | 5.9 |
| Image 1-Manual | 37.9 | 27.4 | 17.5 | 9.5 | 5.1 | 2.7 |

**Table 2.** The rock pixel cumulative area distribution from images 1 to 5, compared with manual results of image 1.

| Image No. | Cumulative Pixel Area Distribution (%) | | | | | |
|---|---|---|---|---|---|---|
| | **2000** | **4000** | **6000** | **8000** | **10,000** | **>10,000** |
| Image 1 | 43.4 | 69.3 | 84.8 | 93.3 | 97.8 | 100 |
| Image 2 | 40.2 | 67.0 | 82.6 | 93.1 | 98.0 | 100 |
| Image 3 | 43.9 | 68.2 | 80.0 | 90.9 | 94.2 | 100 |
| Image 4 | 45.2 | 72.0 | 82.9 | 92.9 | 94.3 | 100 |
| Image 5 | 41.9 | 64.8 | 78.0 | 87.0 | 94.0 | 100 |
| Image 1-Manual | 37.9 | 65.3 | 82.8 | 92.2 | 97.3 | 100 |

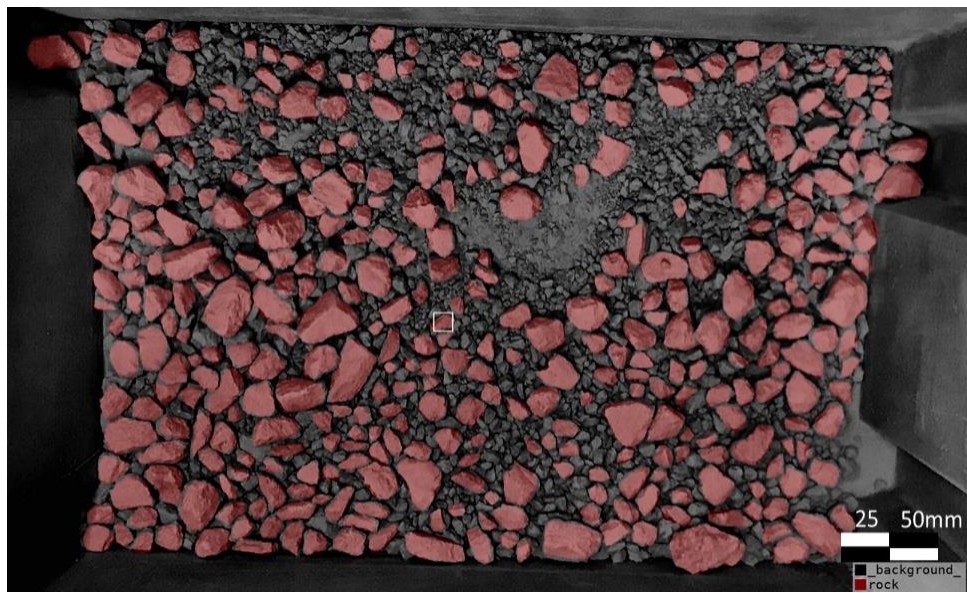

**Figure 10.** Illustration of manual segmentation of image 1. The white rectangle shows the pixel area of 2000.

The cumulative area distribution and their frequencies are shown in Figure 11. The area distribution is found correlated to the edge maps in Figure 7. Such as, in image 4, there are more fines and fewer coarse particles. The fine region can be easily observed in the edge maps, see Figure 9d2, and the fine region is detected by the fine filter. It then results in Figure 11 that image 4 has the finest particles and fewest coarse rocks.

In Tables 3 and 4, the estimated mass distributions from tested images are listed. The laboratory sample is manually screened twice and its particle size distribution is shown in Table 5. Due to shape irregularity, the weight estimation from the pixel area can be biased. In Figure 12a,b, the results of size distributions are presented. The R-squared values are illustrated in Figure 12c. It can be seen from these results that the $R^2 > 0.9$ are obtained for all five images, and their average mass estimation gives $R^2 = 0.975$. It indicates that there is no significant difference between the averaged estimated value to the actual value. In addition, the averaged cumulative distribution shows even higher accuracy with $R^2 = 0.985$.

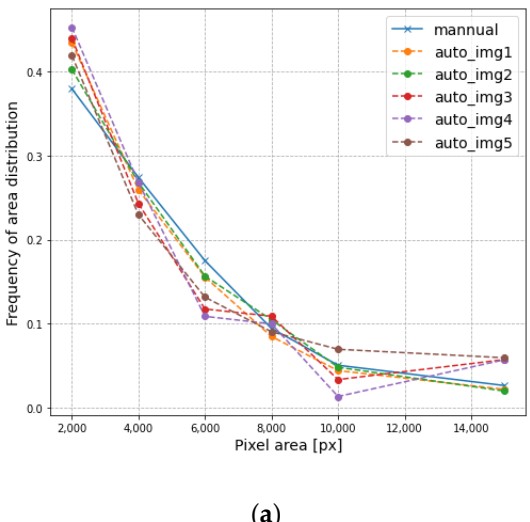

(**a**)

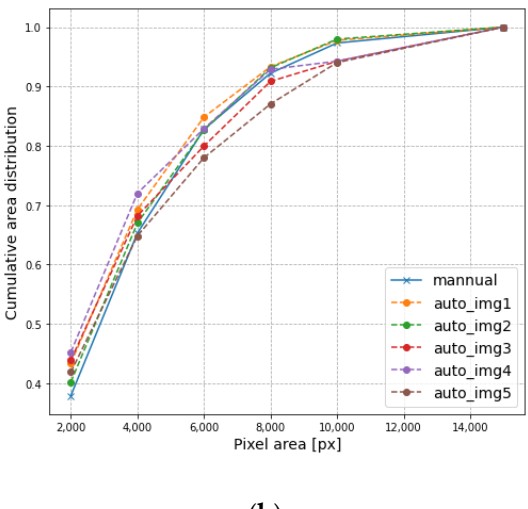

(**b**)

**Figure 11.** The results of the pixel area distribution compared with manual labels of image 1: (**a**) frequency of area distribution; (**b**) cumulative area distribution.

**Table 3.** The frequency of rock mass distribution of images 1 to 5.

| Image No. | Frequency of Mass Distribution (%) | | | | | |
|---|---|---|---|---|---|---|
| | 4 | 8 | 11.2 | 16 | 19 | 22.4 |
| Image 1 | 47.1 | 27.1 | 9.6 | 7.5 | 2.4 | 6.3 |
| Image 2 | 42.3 | 24.2 | 12.4 | 11.8 | 6.0 | 3.2 |
| Image 3 | 42.2 | 22.0 | 10.9 | 12.4 | 5.4 | 7.0 |
| Image 4 | 44.4 | 30.6 | 10.8 | 8.4 | 2.7 | 3.1 |
| Image 5 | 46.8 | 19.3 | 9.9 | 10.5 | 8.7 | 4.7 |

**Table 4.** The cumulative rock mass distribution of images 1 to 5.

| Image No. | Cumulative Mass Distribution (%) | | | | | |
|---|---|---|---|---|---|---|
| | 4 | 8 | 11.2 | 16 | 19 | 22.4 |
| Image 1 | 47.1 | 74.2 | 83.9 | 91.3 | 93.7 | 100 |
| Image 2 | 42.3 | 66.5 | 79.0 | 90.8 | 96.8 | 100 |
| Image 3 | 42.2 | 64.2 | 75.1 | 87.6 | 93.0 | 100 |
| Image 4 | 44.4 | 75.0 | 85.9 | 94.3 | 96.9 | 100 |
| Image 5 | 46.8 | 66.2 | 76.1 | 86.6 | 95.3 | 100 |

**Table 5.** The results of lab sampling.

| Columns | Size (mm) | | | | | |
|---|---|---|---|---|---|---|
| | 4 | 8 | 11.2 | 16 | 19 | 22.4 |
| Weight 1 (g) | 1622.1 | 797 | 521.7 | 400.0 | 234.6 | 194.3 |
| Weight 2 (g) | 1483.0 | 807.1 | 521.6 | 398.5 | 172.6 | 144.7 |
| Frequency of Mass Distribution (%) | 42.6 | 22.0 | 14.3 | 10.9 | 5.6 | 4.6 |
| Cumulative Mass Distribution (%) | 42.6 | 64.5 | 78.8 | 89.8 | 95.4 | 100 |

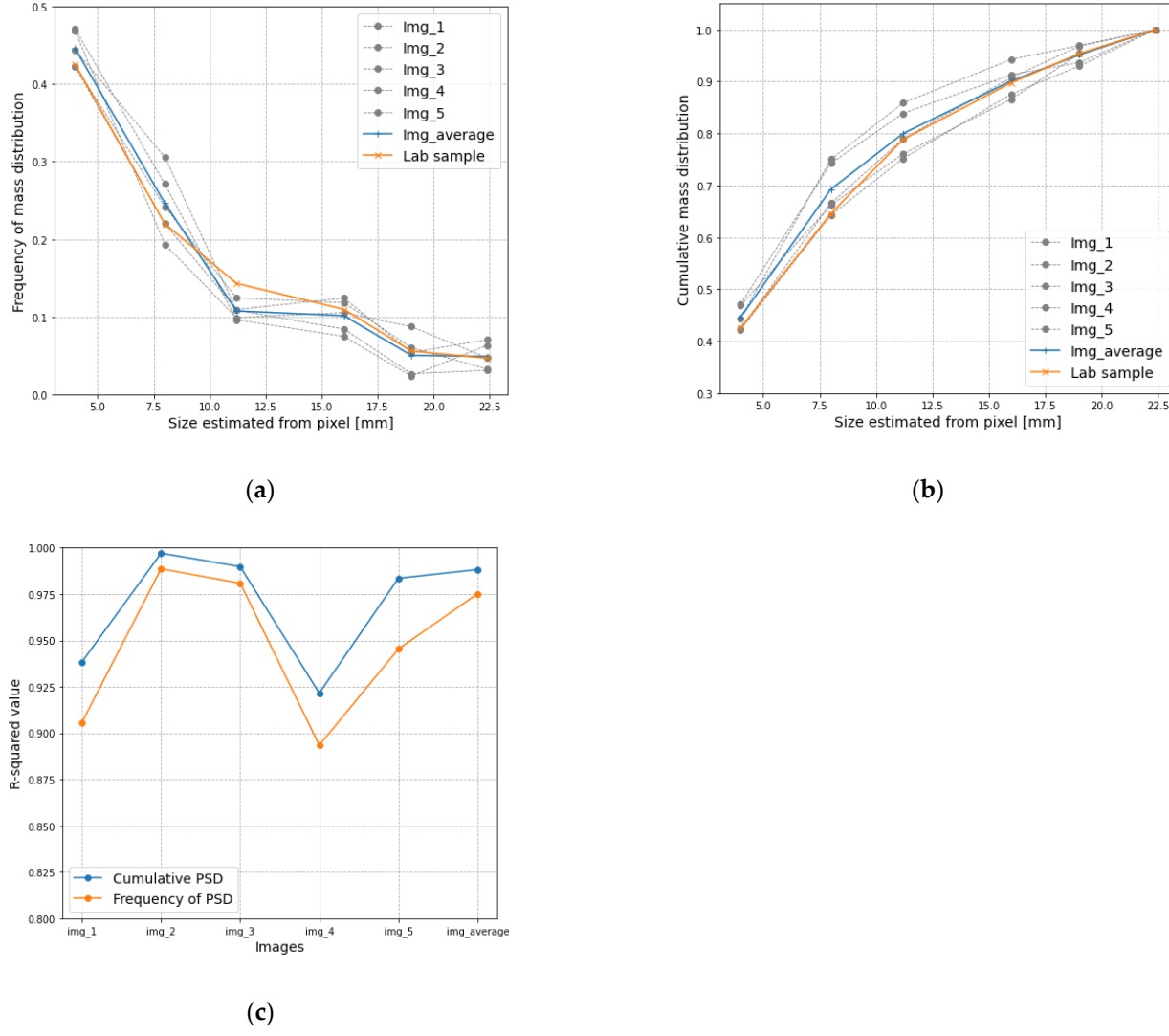

**Figure 12.** The results of the estimated mass distribution compared with lab sampling: (**a**) frequency of the size distribution; (**b**) cumulative size distribution; (**c**) RSME of the estimated results.

The total deviation of the screened samples to the calculated estimation could from (1) wrong edge and contours drawing; (2) area calibration for fine particles; (3) area calibration for erosion loss; (4) pixel area to rock mass conversion; (5) overlapped particles not identified or treated as smaller particles.

The proposed method calculates mass fractions from the non-overlapped rocks. In the experiment, the same sample has been reshuffled five times and similar threshold values are applied for each test. Therefore, the result deviation from (1), (2), (3), (4) is consistent. The differences between these five trials are considered mainly from the overlapped particles. In Table 3, the frequency of the coarse particles varies from 3.1% to 7.0% (−22.4 mm) and from 2.4% to 8.7% (−19 mm). These variances demonstrate that the overlapped particles can introduce considerable bias to the particle size distribution from rock segmentation. It is evident from the averaged results that the deviation caused by overlapped particles can be mitigated by using multiple images from the same sample. A possible industrial solution to the overlapped rocks is installing several cameras at different conveyor positions with different view angles.

## 4. Conclusions

The objective of this study is to develop a method that can estimate the size distributions from the cone crusher product images. The main processes include gray scaling

and adaptive histogram equalization of raw images; edge detection using the modified DexiNed network; contour optimization that filters the fines and connected rocks; the mass estimation from the detected contour areas. The verification and validation are carried out with five representative images of lab samples. It is demonstrated that the proposed method is able to identify the rock edge features and correctly segment coarse material from the connected fine regions and give an accurate estimation of the particle size distribution from the pixel area. Furthermore, this framework of algorithms can be extended to the conveyor belt rock image processing and a wider rock size range.

**Author Contributions:** Conceptualization, H.L., G.A. and M.L.; methodology, H.L.; implementation of the experiments and the writing of the manuscript: H.L.; implementation of the experiments and verification: G.A.; review and editing: G.A. and M.L. All authors have read and agreed to the published version of the manuscript.

**Funding:** This research was partly funded by Innovation Fund Denmark, grant number 5189-00123B.

**Data Availability Statement:** Not applicable.

**Acknowledgments:** The authors thank the anonymous reviewers for their valuable comments to improve the paper quality.

**Conflicts of Interest:** The authors declare no conflict of interest.

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
