# Peer review of "Image Process of Rock Size Distribution Using DexiNed-Based Neural Network"

_minerals, doi:10.3390/min11070736_

Round 1

Reviewer 1 Report

Comments on report

Image Process of Ore Size Distribution Using DexiNed-based Neural Network

Formal:

  1. Headline: neural not nerual
  2. I would rather prefer “particle” size distribution (of rock) to “ore” size distribution, as ore is a restriction to a special type of rock including metal bearing minerals;
  3. 1: “…with sizes ranging from 0 to 22,4 mm” not ranging from 0 – 24 mm
  4. Page 3: paragraph 2.1: Data preparation…; “The sample was shuffled….to get sample images of the same material
  5. “representative” picture: What is a representative picture? Do you mean a representative number of particles or representative for the picture quality…
  6. Telegram style in paragraph 2.3. Segmentation Process; gives the impression of a manual; Full sentences would be more appropriate
  7. articles
  8. a length scale in the photographs would be highly appreciated (although a measure for the box was given in the text. Top 3)
  9. Please give the unit of the figure Rimg. (35 to 39 …..)

Questions:

What was the sample weight of the  -22,4 mm – sample?

Was is it a single layer of the coarse particles in the box ?

Page 5; item 7: What is a “stable” contour map; what is the criterion for stability?

How was the scaling factor Fscale determined? By the weight of single particles?

How is the “box inner size” defined ? How is it related to the pixel size?

What is the relation between pixel size and screen opening size; mesh size?

What is the processing time to obtain a particle size distribution from a picture with calibrated software?

Overall comment

  1. There are a lot of terms used that may call for an explanation for the non image processing community like

“Convex Hull” function, “findContour” function, eroding operator

  1. Please comment the deviation between the average calibrated size distribution from automated image analysis, the manual image analysis and the screen size distribution under the optimum conditions (optimum illumination, no dust, flattened sample,) and with respect a mining application. Would it be necessary to physically sample a stream to obtain a proper picture, or is it possible to find installations for taking pictures on the moving belt, falling stream stockpile?

  1. Please comment the deviation in particle size for calibration (- 22,4 mm) and those streams mentioned in the abstract: feed to an autogenous mill (- 300 mm) and feed to a cone crusher Why did you decide to calibrate the product of a crusher when you think about size distributions much larger in particle size

  1. Ad “accurate estimation of size distribution”. The deviation in the cumulative distributions of 5 times screening of the same sample is for sure much smaller than that compared to 5 times automated image analysis under optimum conditions. Maybe you should use the term “accurate” relative to existing image evaluation routines or for applications, within which the deviation is accurate enough to use the system. Screening of a sample like the one in the paper between 22,4 and 4 mm takes 15 min….

  1. From my point of view it is very interesting to read about the progress in image analysis and automated particle discrimination. From theoretical point of view I would appreciate to have more explanation of the general problems involved in image analysis of particulated samples (reasons why the one or other function is necessary or of benefit); from the practical point of view it would be very interesting whether there is already an industrial application in operation.

Author Response

Thank you for your comments.

Please see the attachment, I have put all comments and replies together.

Best regards,

Haijie

Reviewer 2 Report

Dear Authors, 

Please indicate plant and process-based benefits especially in the introduction and conclusion section. Also, some texts in pictures are hard to read.  

Author Response

(The authors gave the same response as above.)

Reviewer 3 Report

It is an interesting manuscript related to predicting particle size distribution of a cone crusher ore product feeding to mills by using a new kind of CNN-based detection algorithm called DexiNed edge detection network along with contour optimization to detect edge maps of ore images to convert the pixel areas to mass distributions for validating the result by screen analysis, which is widely used but time-consuming technique in mineral processing. Since accurate ore image segmentation is the first step to obtaining reliable particle size information.

Capturing images of the surfaces of piles, which are analyzed by identifying each particle on the surface of the pile and estimating its size of particulate material being transported on conveyor belts using machine vision has been an area of active research for over 25 years. However, there are a number of sources of error relevant to surface-analysis techniques that require consideration and investigation. This manuscript will give a good contribution to the literature.

Paper is of sufficient interest for publication in the journal, title is informative and reflects the content of the paper, abstract is accurate and informative, material in the paper is clearly represented, interpretations and conclusions are sound and justified by data, paper is sufficiently innovative to be published, the work is likely to be used as a source of reference by other workers and the content is original.

But some in order to improve the quality of the manuscript before publish slight revisions could be made upon my comments are below:

  1. Introduction; please give previous literature related to the image analysis studies with the crushed samples for the importance of this study.
  2. Section 2; Start with the materials in the images studied (in P3 “The tested ore images are generated from a laboratory rock sample ………………. proposed method” ) in this Section.
  3. Section 2; In order to use screening results in Table 5 authors should give screening method in detail (such as which sieve standard used? How it was performed?)
  4. Equations 1-5; can be used with their references if they are not original or modified?
  5. For readers to follow the paper better, there are some inconsistent terms used in Table legends and plot legends; while pic1,2…are used in legends of the graphics image 1,2, .. are used in Tables?
  6. P10, 2nd paragraph; Based on Fig 10d; fitting equation along with a R-squared values can be used to evaluate the fitness of the approach to screening results or maximum error percentages between two compared techniques can be discussed.
  7. P10, 2nd paragraph; Above all, what do authors think about this method whether is suitable for ore segmentation with overlapping or not? Overlapping-particle error is due to the fact that many particles are only partially visible, and a large bias toward the smaller size classes results if they are treated as small, entirely visible particles and sized using only their visible profiles. Can this difference be due to overlapped particles since, overlapping-particle error occurs when many particles are only partially visible, which may bias the estimation of particle size because large particles may be treated as smaller particles?
  8. Fig 1; Do the authors think identification of non-overlapped particles, size estimation and weight estimation of particles can be added to the algorithm instead of evaluation?
  9. What is the size limitations of this study since only coarse particles are processed and their sizes are transformed to mass distribution in this study?

Author Response

(The authors gave the same response as above.)
